# Utility of ALT Concentration in Men and Women with Nonalcoholic Fatty Liver Disease: Cohort Study

**DOI:** 10.3390/jcm8040445

**Published:** 2019-04-02

**Authors:** Ki-Chul Sung, Mi-Yeon Lee, Jong-Young Lee, Sung-Ho Lee, Seong-Hwan Kim, Sun H. Kim

**Affiliations:** 1Division of Cardiology, Department of Internal Medicine, Kangbuk Samsung Hospital, Sungkyunkwan University School of Medicine, Seoul 03181, Korea; jyleeheart.lee@samsung.com (J.-Y.L.); shsh96.lee@samsung.com (S.-H.L.); 2Division of Biostatistics, Department of R & D Management, Kangbuk Samsung Hospital, Sungkyunkwan University School of Medicine, Seoul 03181, Korea; my7713.lee@samsung.com; 3Division of Endocrinology, Department of Medicine, Stanford University School of Medicine, Stanford, CA 94305, USA; cardioguy@korea.ac.kr (S.-H.K.); sunhkim@stanford.edu (S.H.K.); 4Stanford Diabetes Research Center, Stanford University School of Medicine, Stanford, CA 94305, USA

**Keywords:** alanine transaminase, non-alcoholic fatty liver disease, fatty liver, insulin resistance, obesity

## Abstract

Nonalcoholic fatty liver disease (NAFLD) is the most common cause of elevated alanine aminotransferase (ALT), but the clinical utility of ALT in detecting and following individuals with NAFLD remains unclear. We conducted a retrospective analysis of 30,988 men and 5204 women with NAFLD diagnosed by ultrasound and stratified them according to sex-specific ALT quartiles. We compared metabolic variables at baseline and repeated ultrasound after at least 6 months among ALT quartiles (Q) in men (Q1 5–24, Q2 25–33, Q3 34–48, Q4 ≥ 49 IU/L) and women (Q1 5–14, Q2 15–20, Q3 21–28, Q4 ≥ 29 IU/L). Prevalence of obesity (BMI ≥ 25 kg/m^2^) and metabolic abnormalities (glucose intolerance, hypertension) significantly (*p* < 0.001) increased from ALT Q1 to Q4 in both men and women at baseline. After a mean follow-up of 4.93 years, 17.6% of men and 31.1% of women resolved their NAFLD. The odds ratio (OR) of resolving significantly (*p* < 0.001) decreased by quartiles even after multiple adjustments. The adjusted OR for resolution in Q4 was 0.20 (0.18–0.23) in men and 0.35 (0.26–0.47) in women compared with Q1. Individuals with NAFLD span the full range of ALT concentrations, but those with the highest ALT have the worst metabolic profile and persistent NAFLD.

## 1. Introduction

Alanine aminotransferase (ALT) is the most widely used biomarker to detect liver disease in clinical practice. The upper limit of normal (ULN) of ALT varies by laboratories and can range widely from 31 to 72 IU/L [1]. One reason for this variability may relate to the reference population used to set the ULN and the inadvertent inclusion of people with nonalcoholic fatty liver disease (NAFLD) [2].

NAFLD is currently the most common cause of elevated ALT worldwide [3]. Risk for NAFLD increases with insulin resistance and metabolic risk factors, including obesity, dysglycemia, and dyslipidemia [4]. When reference populations have been created to minimize the inclusion of individuals with these risk factors, the ULN of ALT has been lower: down to 30 for men and 19 IU/L for women [5]. While better delineating ULN, there have been concerns that lowering the ULN would increase the proportion with ALT elevation and increase unnecessary medical evaluation and cost [6]. For example, applying these ULN to a National Health and Nutrition Examination Survey cohort would classify 36% of U.S. adults as having abnormal ALT [7].

Similar objections have been raised to lowering the cut point for fasting glucose [8,9] and blood pressure [10]. On the other hand, glucose concentration and blood pressure limits are used as the criterion to diagnose their respective diseases—diabetes and hypertension—whereas an ALT concentration is not used to define a specific disease. Thus, lowering the ULN of ALT may increase the inclusion of individuals with liver disease, but having a normal ALT does not assure an absence of liver disease.

To understand the clinical role of ALT concentration in NAFLD, we compared adults with ultrasound-diagnosed NAFLD stratified by sex-specific quartiles of ALT. We also evaluated baseline ALT in determining rates of regression or persistence of fatty liver at follow-up.

## 2. Materials and Methods

This was a retrospective cohort study to evaluate the utility of ALT concentration in South Korean adults with a fatty liver diagnosed based on ultrasound. The study cohort consisted of individuals who participated in a comprehensive health-screening program, at least twice, at Kangbuk Samsung Hospital, Seoul and Suwon, Korea from 2002 to 2014 (*n* = 259,011). Most individuals (over 80%) were employees of various companies or local government organizations, or the spouses of these employees. In South Korea, the Industrial Safety and Health Law requires annual or biennial health screening examinations of all employees, which is offered free of charge. Individuals were excluded from the study if they were less than 20 years old and consumed >30 g/day (men) or >20 g/day (women) of alcohol [11]. We also excluded individuals who were receiving treatment for diabetes, hypertension, or hyperlipidemia (including statin treatment). Individuals were also excluded for potential secondary etiologies of liver disease: positive hepatitis C antibody status (*n* = 69); positive hepatitis B surface antigen status (*n* = 1753); and evidence of cancer (*n* = 1675). Individuals were also excluded for missing data (*n* = 11,299 at baseline and *n* = 12,642 at follow up). After exclusions, there were 158,638 total individuals, and 36,192 (22.8%) had a fatty liver detected by ultrasound. The mean follow-up was 4.93 years (±3.39), and the median was 3.94 years (maximum 12.65 years). The study was approved by the Institutional Review Board of Kangbuk Samsung Hospital (KBSMC 2013-01-010) and any requirement for informed consent was waived by the Board, because de-identified information was retrieved retrospectively.

### 2.1. Measurements

As part of the health-screening program, individuals completed self-administered questionnaires, related to their medical and social history and medication use. All methods were performed in accordance with the relevant guidelines and regulations. Individuals were asked about duration of education (years) and smoking history (never, former, or current). Individuals were also asked about their frequency of alcohol intake and quantity of intake when drinking; this information was used to calculate alcohol intake per day (grams/day). We also assessed the frequency of moderate- or vigorous-intensity physical activity per week by asking how many days per week they participated in a sweat-inducing exercise for more than 30 min.

Trained staff collected anthropometric measurements and vital statistics. Body weight was measured in light clothing with no shoes to the nearest 0.1 kg using a digital scale. Height was measured to the nearest 0.1 cm. Body mass index (BMI) was calculated as weight in kilograms divided by height in meters squared. Obesity was defined as a BMI ≥ 25 kg/m^2^. Waist circumference measurements were not evaluated as they were missing at baseline in 40% of individuals.

Blood samples were collected after at least 10 h of fasting and analyzed in the same core clinical laboratory at Kangbuk Samsung Hospital. The core clinical laboratory has been accredited and participates annually in inspections and surveys by the Korean Association of Quality Assurance for Clinical Laboratories. Insulin concentrations were available for 13,688 individuals (38%). Serum glucose, low-density lipoprotein cholesterol (LDL-C), high-density lipoprotein cholesterol (HDL-C) and triglyceride concentrations were measured using Bayer Reagent Packs (Bayer Diagnostics, Leverkusen, Germany) on an automated chemistry analyzer (Advia 1650 Autoanalyzer; Bayer Diagnostics). High sensitivity C-reactive protein (hs-CRP) was analyzed using particle-enhanced immunonephelometry on a BNIITM System (Dade Behring, Marburg, Germany). The homeostatic model assessment of insulin resistance (HOMA-IR) was used as a surrogate measure of insulin resistance and calculated by using the following equation: $\frac{fasting\;insulin\left(\frac{mU}{L}\right)xfasting\;glucose\left(\frac{mmol}{L}\right)}{22.5}$ [12].

Clinical radiologists performed abdominal ultrasonography (Logic Q700 MR; GE, Milwaukee, WI, USA) using a 3.5 MHz probe for all participants at baseline and follow-up. The following images were undertaken: (i) sagittal view of the right lobe of the liver and right kidney, (ii) transverse view of the left lateral segment of the liver and spleen, and (iii) transverse view of the liver for altered echo texture. Individuals were classified as having a fatty liver if there was an increase in the echogenicity of the liver compared with the echogenicity of the renal cortex where the diaphragm and intrahepatic vessels appeared normal. Individuals diagnosed with a fatty liver were provided a handout about the association between a fatty liver and obesity. They were also advised to get regular exercise, maintain a normal weight, reduce caloric intake, eat moderate carbohydrates, and limit added sugars, fried foods and alcohol to reduce fat in the liver.

### 2.2. Statistical Analyses

The statistical analysis was performed using STATA version 15.0 (StataCorp LP, College Station, TX, USA). Reported *p*-values were two-tailed, and <0.05 was considered statistically significant. Normal distribution of continuous variables was evaluated using graphical methods to check for skewness and kurtosis. Nonparametric variables (insulin, triglyceride, AST, ALT, hs-CRP) are reported as median (interquartile range, IQR) and logarithmically transformed before analyses. We used receiver operating characteristic (ROC) curves to find the optimal ALT cut points to identify ultrasound-diagnosed fatty liver in men and women using the Youden index. We also stratified the study population with a fatty liver by sex-specific quartiles of ALT, and comparisons were made using one-way ANOVA. We conducted multiple linear regression analyses to evaluate the effect of ALT quartile on the resolution of fatty liver at follow-up. Analyses were adjusted for age, baseline BMI, BMI change, education, exercise, smoking and alcohol intake.

## 3. Results

After inclusion and exclusion criteria were applied, we identified 158,638 total individuals, and 22.8% (36,192) had a fatty liver detected by ultrasound. More men than women had a fatty liver (34.3% vs. 7.6%). An ALT cut point of 27 in men and 19 in women showed the highest sensitivity (71% and 60%) and specificity (72% and 77%) with an area under the ROC curve of 78% and 75%, respectively.

Figure 1 shows the number of individuals with a fatty liver relative to the total population by sex-specific ALT quartiles in men (a) and women (b). ALT quartiles were determined in those with a fatty liver, and had wider ranges and higher concentrations in men than women. In both men and women, the proportion with a fatty liver relative to the total population (with and without fatty liver) increased from Q1 to Q4. Interestingly, the majority of men (77.5%) in the highest quartile had a fatty liver vs. 32% of women in Q4.

Table 1 and Table 2 show baseline characteristics by sex-specific ALT quartiles. In men, age significantly declined from ALT Q1 to Q4. In contrast, BMI increased and the percentage with obesity (BMI ≥ 25 kg/m^2^) increased from 45.8% in Q1 to 75.3% in Q4. Also, men in the higher ALT quartiles were more likely to attain higher education (>high school), exercise less, and be current smokers. Higher ALT concentrations were also associated with insulin resistance and worse metabolic profile (higher HOMA-IR, glucose, insulin, triglyceride concentration, and lower HDL-C concentration). Consequently, individuals in the highest ALT quartile had a higher hs-CRP and greater prevalence of glucose intolerance (prediabetes and diabetes) and hypertension.

In contrast to men, age increased from ALT Q1 to Q4 in women. Also, women in the higher ALT quartile were less likely to attain higher education but more likely to exercise. Obesity and metabolic changes associated with ALT concentrations were similar to men, but more pronounced in women. For example, women with a fatty liver in Q4 were 5-fold more likely to have diabetes compared with those in Q1; in men, it was less than a 2-fold increase. Similarly, women in Q4 were 3-fold more likely to have hypertension compared with the 1.5-fold increase in men.

After a mean follow up of 4.93 years, 17.6% of men and 31.1% of women resolved their fatty liver. As seen in Figure 2, the likelihood of resolving declined significantly as ALT increased for both men (a) and women (b). In men, only 9.6% of those in the highest ALT quartile were likely to resolve compared with 26.1% in the lowest quartile. In women, 22.1% resolved in the highest quartile compared with 43.8% in the lowest. Thus, adjusted OR for resolution was low in both men and women (0.20 and 0.35, respectively) in ALT Q4.

Factors associated with the resolution of fatty liver by ultrasound are shown in Table 3 and Table 4 by sex-specific ALT quartiles in men and women, respectively. Regardless of quartiles, factors associated with resolution were similar; however, the magnitude of difference between those who resolved versus persisted were greater as ALT levels increased. For example, starting with men, a decline in BMI was associated with fatty liver resolution regardless of ALT quartile, but the difference in BMI change was greater in Q4 than Q1 (−1.75 vs. −0.59). Those who resolved also had a greater decrease in triglyceride and an increase in HDL-C concentration. Glucose tended to increase in all groups, but the increase was attenuated in those who resolved their fatty liver. There was also an attenuated increase in LDL-C concentration. There was a significant differential in the AST and ALT change between those who resolved versus persisted. In Q3 and Q4, AST and ALT concentrations tended to decrease in all groups, but the decrease was greater in those who resolved their fatty liver at follow-up than those who persisted. Factors associated with the resolution of fatty liver in women were similar to men. Thus, those who resolved had a decreased BMI and triglyceride and increase in HDL-C concentration. Glucose, LDL-C, AST and ALT were also likely to decline in those who resolved their fatty liver, especially in those in Q4. In women as compared with men, hs-CRP were also likely to decline in those who resolved, and the difference was significant in Q3 and Q4. In men, hs-CRP changes were not significantly different in those who resolved or persisted with their fatty liver.

## 4. Discussion

Recently, there has been increasing discussion regarding lowering the ULN of ALT [6,13,14]. The suggested ULN for ALT has generally been ≤35 for men and ≤26 for women, when defined using healthy populations with low risk for NAFLD and normal BMI, triglyceride and glucose concentrations [5,15,16,17]. Studies have also excluded people with steatosis (>5%) based on liver biopsy [15] or ultrasound [16,17]. In our population with an ultrasound-diagnosed fatty liver, optimal ALT cut points were lower than previously suggested, and half (those in quartile 1 and 2) had ALT <34 in men and <21 in women, which are below the ULN values suggested for men and women, respectively. Thus, a significant number of people have a fatty liver with apparently normal ALT concentrations.

While a sizeable number of people with a fatty liver have normal ALT concentrations, the proportion with a fatty liver increases as ALT concentration increases (Figure 1). This increase was striking in men (Figure 1a), and the majority (77.5%) in Q4 had a fatty liver diagnosed by ultrasound. In women (Figure 1b), the increase with ALT quartiles was more modest, with 32% having a fatty liver in Q4. Sex-differences in ALT concentrations are well known [5], with women having lower concentrations than men. Sex differences in NAFLD prevalence are complex and may vary with age, sex hormones, and alcohol-use patterns [18]. Compared with men, women are believed to have a later peak in prevalence of NAFLD, which occurs after the age of 50 years or after the onset of menopause [19,20]. Our population was relatively young and likely premenopausal, and age may be one factor explaining the sex differential in the prevalence of fatty liver. This may also explain why age increased by ALT quartiles in women but declined in men.

Despite differences in age and the prevalence of fatty liver, metabolic factors associated with an increase in ALT quartiles were similar between men and women, and perhaps more pronounced in women. Thus, regardless of sex, individuals with higher ALT concentrations had higher HOMA-IR and glucose, insulin, and triglyceride concentrations and lower HDL-C concentration—all known biomarkers of insulin resistance [21,22]. The relative increase in diabetes and hypertension in Q4 compared with Q1 were higher in women than men. While this may relate to demographic differences (such as age), it may also suggest that the prevalence of fatty liver in women (especially premenopausal women) may identify those with the highest risk for metabolic diseases. In general, the impact of insulin resistance is attenuated in premenopausal versus postmenopausal women [21] and may explain some of the female advantage [23] in cardiovascular risk. Thus, younger women with fatty liver may represent the subset without this metabolic advantage.

Finally, individuals with higher ALT concentrations were less likely to resolve their fatty liver at follow up (Figure 2), and men were less likely to resolve than women. Factors associated with resolution were similar regardless of ALT quartile; however, the magnitude of change needed to resolve was greater at higher ALT levels. For example, weight loss was universally associated with resolution of fatty liver regardless of baseline ALT levels. However, in Q4, the differential in BMI change between those who resolved versus persisted was greater than in Q1. This was also true for changes in triglyceride, HDL-C, LDL-C and glucose concentration, as well as changes in ALT and AST concentration.

There are limitations to our study. We had more men than women; thus, our findings may be less generalizable to women than men. This was a retrospective cohort study, and hence we can only report associations and not causations between ALT and NAFLD prevalence and resolution. Finally, we diagnosed fatty liver based on ultrasound. Ultrasound has been shown to be reliable and accurate for detecting moderate-to-severe fatty liver [24]. However, ultrasound may miss milder degrees of fatty liver, when steatosis is less than 20% [25]. Nevertheless, ultrasound provides a noninvasive and feasible means to evaluate fatty liver in a large population, and is currently recommended as the first-line diagnostic procedure for the imaging of NAFLD [11].

## 5. Conclusions

In conclusion, ALT is an important biomarker for fatty liver disease but care needs to be taken in defining the ULN. Unlike glucose concentration and blood pressure, ALT levels do not define a specific disease. As seen in this study, 50% of individuals with a known fatty liver can have ALT <34 in men and <21 in women—a lower level for ALT ULN than the majority of labs in the United States [1,2]. Thus, individuals with a fatty liver can span the full range of ALT concentrations, and goals for setting the ULN needs to be better delineated. In men, an ALT level ≥49 IU/L was present in only 11%, but 3 out 4 men had a fatty liver diagnosed by ultrasound. This group also had the worst metabolic profile and lowest likelihood to resolve their fatty liver. In women, although the prevalence was less, those in the upper ALT quartile (ALT ≥ 29) with a fatty liver had a relatively greater increase in metabolic diseases including diabetes and hypertension. While our study needs to be interpreted with caution given its retrospective design, we suggest that a higher ULN may yield better case findings for those with fatty livers who deserve the greatest clinical attention.

## Figures and Tables

**Figure 1 jcm-08-00445-f001:**
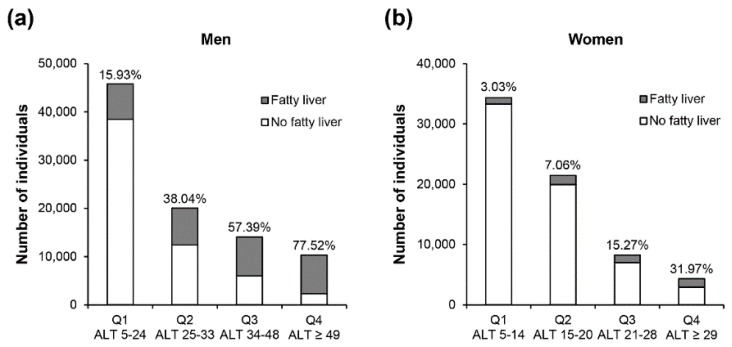
Proportion with and without a fatty liver in the total population by ALT concentration in men (**a**) and women (**b**). ALT = alanine aminotransferase.

**Figure 2 jcm-08-00445-f002:**
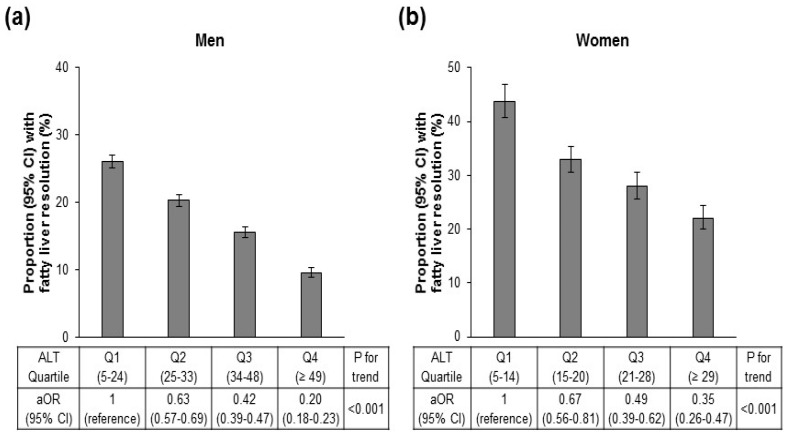
Odds ratio (95% confidence interval, CI) for fatty liver resolution by ALT quartiles in men (**a**) and women (**b**). OR is adjusted (aOR) for age, sex, baseline BMI, BMI change, education, exercise, smoking and alcohol intake (g/day). ALT = alanine aminotransferase, BMI = body mass index.

**Table 1 jcm-08-00445-t001:** Baseline characteristics by ALT quartiles in men with a fatty liver.

ALT Range	Q 1 5–24 IU/L (*n* = 7294)	Q2 25–33 IU/L (*n* = 7633)	Q3 34–48 IU/L (*n* = 8062)	Q4 ≥49 IU/L (*n* = 7999)	*p* for Trend
Age (years)	37.3 ± 7.3	37.0 ± 7.0	36.3 ± 6.4	35.2 ± 5.6	<0.001
BMI (kg/m^2^)	24.9 ± 2.3	25.6 ± 2.4	26.1 ± 2.5	27.0 ± 2.8	<0.001
Education, *n* (%)					<0.001
≤High school	481 (6.6)	589 (7.7)	592 (7.3)	657 (8.2)	
>High school	4309 (59.1)	4606 (60.3)	5089 (63.1)	4977 (62.2)	
Unknown	2054 (34.3)	2438 (32.0)	2381 (29.5)	2365 (29.6)	
Exercise, *n* (%)					<0.001
<1 time per week	3616 (49.6)	3931 (51.5)	4380 (54.3)	4594 (57.4)	
≥1 time per week	3678 (50.4)	3702 (48.5)	3682 (45.7)	3405 (42.6)	
Smoking, *n* (%)					<0.001
Never/former	4367 (59.9)	4515 (59.2)	4615 (57.2)	4446 (55.6)	
Current	2809 (38.5)	3023 (39.6)	3354 (41.6)	3481 (43.5)	
Unknown	118 (1.6)	95 (1.2)	93 (1.2)	72 (0.9)	
Glucose (mg/dL)	95.5 ± 12.4	95.6 ± 11.3	95.9 ± 12.4	97.0 ± 12.2	<0.001
Insulin (µIU/mL) ^1^	5.7 (4.1, 7.7)	6.4 (4.7, 8.7)	7.2 (5.2, 9.7)	8.6 (6.1, 11.6)	<0.001
HOMA-IR ^1^	1.3 (0.9, 1.8)	1.5 (1.1, 2.1)	1.7 (1.2, 2.4)	2.0 (1.4, 2.8)	<0.001
LDL-C (mg/dL)	122.8 ± 28.3	126.4 ± 28.3	129.7 ± 29.3	135.1±30.2	<0.001
Triglyceride (mg/dL)	130 (96, 177)	147 (108, 201)	156.5 (115, 214)	172 (126, 232)	<0.001
HDL-C (mg/dL)	49.2 ± 9.9	48.3 ± 9.2	48.0 ± 9.2	46.9 ± 8.7	<0.001
AST (IU/L)	20 (17, 22)	23 (21, 26)	27 (24, 30)	37 (32, 45)	<0.001
ALT (IU/L)	20 (17, 22)	29 (27, 31)	40 (36, 44)	65(55, 84)	<0.001
hs-CRP (mg/dL)	0.06 (0.03, 0.11)	0.07 (0.04, 0.12)	0.07 (0.04, 0.14)	0.09 (0.05, 0.16)	<0.001
Obese, *n* (%)	3342 (45.8)	4355 (57.1)	5270 (65.4)	6021 (75.3)	<0.001
Prediabetes, *n* (%)	1823 (25.0)	2059 (26.9)	2170 (26.9)	2504 (31.3)	<0.001
Diabetes, *n* (%)	140 (2.5)	120 (2.3)	169 (3.3)	227 (4.6)	<0.001
Hypertension, *n* (%)	806 (11.1)	1052 (13.8)	1118 (14.8)	1418 (17.7)	<0.001
CVD, *n* (%)	188 (2.6)	193 (2.5)	200 (2.5)	215 (2.7)	0.71

^1^ Median (interquartile). ALT = alanine aminotransferase, AST = aspartate transaminase, BMI = body mass index, CVD = cardiovascular disease, HDL = high-density lipoprotein cholesterol, HOMA-IR = homeostatic model assessment of insulin resistance, hs-CRP = high-sensitivity c-reactive protein, LDL = low-density lipoprotein cholesterol. Obese was defined as having BMI ≥ 25 kg/m^2^; prediabetes, fasting glucose 100–125 mg/dL; diabetes, fasting glucose ≥ 126 mg/dL, hypertension, systolic blood pressure ≥ 140 mmHg and/or diastolic blood pressure ≥ 90 mmHg; CVD, self-reported coronary artery disease or stroke.

**Table 2 jcm-08-00445-t002:** Baseline characteristics by ALT quartiles in women with a fatty liver.

ALT range	Q1 5–14 IU/L (*n* = 1042)	Q2 15–20 IU/L (*n* = 1516)	Q3 21–28 IU/L (*n* = 1262)	Q4 ≥29 IU/L (*n* = 1384)	*p* for Trend
Age (years)	37.2 ± 6.9	38.3 ± 8.1	40.4 ± 9.4	39.2 ± 8.9	<0.001
BMI (kg/m^2^)	24.2 ± 2.9	25.1 ± 3.2	25.6 ± 3.1	26.6 ± 3.4	<0.001
Education, *n* (%)					<0.001
≤High school	213 (20.4)	368 (24.2)	359 (28.5)	415 (30.0)	
>High school	499 (47.9)	633 (41.8)	507 (40.2)	516 (37.3)	
Unknown	330 (31.7)	515 (34.0)	396 (31.4)	453 (32.7)	
Exercise, *n* (%)					0.049
<1 time per week	718 (68.9)	1016 (67.0)	847 (67.1)	898 (64.9)	
≥1 time per week	324 (31.1)	500 (33.0)	415 (32.9)	486 (35.1)	
Smoking, *n* (%)					0.067
Never/former	959 (92.0)	1413 (93.2)	1184 (93.8)	1301 (94.0)	
Current	28 (2.7)	45 (3.0)	27 (2.1)	39 (2.8)	
Unknown	55 (5.3)	58 (3.8)	51 (4.1)	44 (3.2)	
Glucose (mg/dL)	94.1 ± 12.3	94.9 ± 10.7	96.5 ± 16.6	99.1 ± 21.2	<0.001
Insulin (µIU/mL) ^1^	6.1 (4.4, 8.9)	7.6 (5.3, 10.5)	8.3 (5.9, 11.3)	10.1 (6.7, 14.5)	<0.001
HOMA-IR ^1^	1.4 (1.0, 2.1)	1.8 (1.2, 2.5)	2.0 (1.4, 2.7)	2.4 (1.6, 3.5)	<0.001
LDL-C (mg/dL)	116.4 ± 29.4	120.3 ± 29.3	125.7 ± 30.7	128.9 ± 32.3	<0.001
Triglyceride (mg/dL)	97 (72, 133)	113 (80, 156)	122 (91, 169)	137 (98, 190)	<0.001
HDL-C (mg/dL)	55.5 ± 12.3	54.3 ± 11.7	53.6 ± 11.1	52.3 ± 11.2	<0.001
AST (IU/L)	16 (14, 18)	19 (17, 21)	22 (20, 25)	30 (25, 36)	<0.001
ALT (IU/L)	12 (11, 13)	17 (16, 19)	24 (22, 26)	39 (33, 52)	<0.001
hs-CRP (mg/dL)	0.07 (0.03, 0.15)	0.07 (0.04, 0.16)	0.09 (0.05, 0.18)	0.11 (0.06, 0.22)	<0.001
Obese, *n* (%)	359 (34.5)	689 (45.5)	666 (52.3)	905 (65.4)	<0.001
Prediabetes, *n* (%)	200 (19.2)	384 (25.3)	345 (27.3)	417 (30.1)	<0.001
Diabetes, *n* (%)	14 (1.7)	29 (2.7)	47 (5.6)	90 (9.9)	<0.001
Hypertension, *n* (%)	36 (3.5)	110 (7.3)	124 (9.8)	145 (10.5)	<0.001
CVD, *n* (%)	19 (1.8)	62 (4.1)	35 (2.8)	56 (4.1)	0.037

ALT = alanine aminotransferase, AST = aspartate transaminase, BMI = body mass index, CVD = cardiovascular disease, HDL-C = high-density lipoprotein cholesterol, HOMA-IR = homeostatic model assessment of insulin resistance, hs-CRP = high-sensitivity c-reactive protein, LDL-C = low-density lipoprotein cholesterol. Obese was defined as having BMI ≥ 25 kg/m^2^; prediabetes, fasting glucose 100–125 mg/dL; diabetes, fasting glucose ≥ 126 mg/dL, hypertension, systolic blood pressure ≥ 140 mmHg and/or diastolic blood pressure ≥ 90 mmHg; CVD, self-reported coronary artery disease or stroke. ^1^ Median (interquartile).

**Table 3 jcm-08-00445-t003:** Prospective change in key metabolic factors based on ALT quartiles and resolution of fatty liver in men.

	Q1 5–24 IU/L (*n* = 7294)		Q2 25–33 IU/L (*n* = 7633)		Q3 34–48 IU/L (*n* = 8062)		Q4 ≥ 49 IU/L (*n* = 7999)	
	Resolved (*n* = 1901)	Persisted (*n* = 5393)	Resolved (*n* = 1548)	Persisted (*n* = 6085)	Resolved (*n* = 1253)	Persisted (*n* = 6809)	Resolved (*n* = 767)	Persisted (*n* = 7232)
BMI change	−0.59 ± 1.18 (−0.64, −0.54)	0.40 ± 1.10 ^1^ (0.37, 0.43)	−0.98 ± 1.22 (−1.04, −0.92)	0.27 ± 1.10 ^1^ (0.24, 0.3)	−1.29 ± 1.49 (−1.37, −1.21)	0.14 ± 1.14 ^1^ (0.11, 0.16)	−1.75 ± 1.66 (−1.87, −1.63)	0.02 ± 1.27 ^1^ (−0.01, 0.05)
Glucose change	0.97 ± 12.75 (0.39, 1.54)	2.23 ± 11.38 ^1^ (1.93, 2.54)	0.48 ± 10.01 (−0.02, 0.98)	2.96 ± 11.06 ^1^ (2.69, 3.24)	2.05 ± 12.96 (1.33, 2.77)	3.66 ± 13.5 ^1^ (3.34, 3.98)	1.20 ± 16.19 (0.06, 2.35)	5.13 ± 18.06 ^1^ (4.71, 5.54)
LDL-C change	4.17 ± 23.21 (3.11, 5.22)	9.34 ± 22.42 ^1^ (8.74, 9.94)	1.07 ± 24.21 (−0.14, 2.29)	7.97 ± 23.81 ^1^ (7.37, 8.58)	−0.56 ± 26.28 (−2.03, 0.92)	6.09 ± 24.09 ^1^ (5.52, 6.67)	−4.59 ± 28.49 (−6.62, −2.55)	2.46 ± 24.51 ^1^ (1.89, 3.03)
TG change	−13.91 ± 74.08 (−17.24, −10.58)	11.33 ± 81.03 ^1^ (9.17, 13.5)	−25.51 ± 74.97 (−29.24, −21.77)	7.66 ± 84.49 ^1^ (5.53, 9.78)	−34.92 ± 85.49 (−39.66, −30.18)	1.21 ± 91.06 ^1^ (−0.96, 3.37)	−49.66 ± 88.74 (−55.95, −43.37)	−4.03 ± 98.31 ^1^ (−6.3, −1.77)
HDL-C change	2.50 ± 8.68 (2.11, 2.89)	−0.66 ± 7.38 ^1^ (−0.86, −0.46)	3.07 ± 8.78 (2.63, 3.5)	−0.86 ± 7.49 ^1^ (−1.05, −0.68)	2.94 ± 9.26 (2.42, 3.45)	−0.9 ± 7.42 ^1^ (−1.08, −0.73)	3.17 ± 9.15 (2.52, 3.82)	−1.1 ± 7.42 ^1^ (−1.27, −0.93)
AST change	0.20 ± 7.45 (−0.13, 0.54)	2.59 ± 9.97 ^1^ (2.32, 2.85)	−1.74 ± 16.11 (−2.55, −0.94)	1.49 ± 18.35 ^1^ (1.02, 1.95)	−5.44 ± 10.18 (−6, −4.88)	−0.5 ± 12.54 ^1^ (−0.8, −0.2)	−15.53 ± 22.19 (−17.1, −13.96)	−7.66 ± 20.84 ^1^ (−8.14, −7.18)
ALT change	0.34 ± 8.82 (−0.06, 0.73)	6.47 ± 14.68 ^1^ (6.08, 6.86)	−5.97 ± 9.48 (−6.44, −5.5)	3.80 ± 17.54 ^1^ (3.36, 4.24)	−13.72 ± 13.67 (−14.48, −12.96)	−0.84 ± 21.05 ^1^ (−1.34, −0.34)	−38.29 ± 35.23 (−40.78, −35.79)	−20.12 ± 36.38 ^1^ (−20.96, −19.28)
hs-CRP change	0.015 ± 0.52 (−0.02, 0.05)	0.012 ± 0.4 (−0.002, 0.03)	−0.010 ± 0.33 (−0.03, 0.01)	0.001 ± 0.44 (−0.01, 0.02)	0.042 ± 0.88 (−0.03, 0.11)	0.006 ± 0.38 (−0.01, 0.02)	−0.011 ± 0.51 (−0.06, 0.04)	0.001 ± 0.37 (−0.01, 0.01)

Data are mean ± standard deviation. Change was calculated as BMI_follow-up_−BMI_baseline_. ^1^
*p* < 0.05 between individuals who resolved and persisted using independent *t*-tests. ALT = alanine aminotransferase, AST = aspartate transaminase, BMI = body mass index, HDL-C = high-density lipoprotein cholesterol, hs-CRP = high-sensitivity c-reactive protein, LDL-C = low-density lipoprotein cholesterol, TG = triglyceride.

**Table 4 jcm-08-00445-t004:** Prospective change in key metabolic factors based on ALT quartiles and resolution of fatty liver in women.

	Q1 5–14 IU/L (*n* = 1042)		Q2 15–20 IU/L (*n* = 1516)		Q3 21–28 IU/L (*n* = 1262)		Q4 ≥29 IU/L (*n* = 1384)	
	Resolved (*n* = 456)	Persisted (*n* = 586)	Resolved (*n* = 500)	Persisted (*n* = 1016)	Resolved (*n* = 354)	Persisted (*n* = 908)	Resolved (*n* = 306)	Persisted (*n* = 1078)
BMI change	−0.62 ± 1.46 (−0.75, −0.49)	0.67 ± 1.44 ^1^ (0.55, 0.79)	−0.99 ± 1.5 (−1.12, −0.86)	0.38 ± 1.38 ^1^ (0.29, 0.46)	−1.29 ± 1.54 (−1.45, −1.13)	0.25 ± 1.42 ^1^ (0.15, 0.34)	−1.85 ± 2.15 (−2.09, −1.61)	0.07 ± 1.57 ^1^ (−0.03, 0.16)
Glucose change	−1.17 ± 8.04 (−1.91, −0.43)	1.66 ± 9.57 ^1^ (0.88, 2.43)	−0.27 ± 9.05 (−1.07, 0.52)	3.08 ± 12.38 ^1^ (2.32, 3.84)	−0.59 ± 11.07 (−1.74, 0.57)	4.26 ± 18.85 ^1^ (3.03, 5.49)	−2.36 ± 12.07 (−3.72, −1)	5.63 ± 23.87 ^1^ (4.21, 7.06)
LDL-C change	1.94 ± 21.38 (−0.04, 3.91)	9.72 ± 24.35 ^1^ (7.74, 11.71)	1.64 ± 25.71 (−0.62, 3.91)	8.52 ± 24.31 ^1^ (7.01, 10.03)	−1.38 ± 28.18 (−4.37, 1.6)	6.31 ± 27.10 ^1^ (4.52, 8.1)	−6.18 ± 28.89 (−9.47, −2.89)	4.18 ± 27.75 ^1^ (2.49, 5.86)
TG change	−7.33 ± 50.23 (−11.95, −2.7)	10.96 ± 63.28 ^1^ (5.83, 16.1)	−16.35 ± 59.19 (−21.55, −11.15)	6.37 ± 78.72 ^1^ (1.52, 11.22)	−28.11 ± 77.19 (−36.18, −20.04)	2.35 ± 75.83 ^1^ (−2.59, 7.28)	−31.98 ± 61.83 ^1^ (−38.93, −25.02)	−4.7 ± 88.72 ^1^ (−10, 0.6)
HDL-C change	4.27 ± 10.57 (3.3, 5.24)	−0.18 ± 9.34 ^1^ (−0.93, 0.58)	3.69 ± 11.11 (2.72, 4.67)	−0.52 ± 8.98 ^1^ (−1.07, 0.04)	2.60 ± 11.8 (1.37, 3.83)	−1.16 ± 9.39 ^1^ (−1.77, −0.55)	4.13 ± 12.23 ^1^ (2.76, 5.51)	−0.42 ± 9.59 ^1^ (−1, 0.15)
AST change	0.95 ± 5.22 (0.46, 1.43)	3.12 ± 7.96 ^1^ (2.48, 3.77)	−0.51 ± 4.93 (−0.94, −0.08)	2.80 ± 14.94 ^1^ (1.88, 3.72)	−2.73 ± 8.32 (−3.6, −1.86)	1.45 ± 10.96 ^1^ (0.74, 2.16)	−11.28 ± 18.95 (−13.41, −9.15)	−4.30 ± 21.00 ^1^ (−5.56, −3.05)
ALT change	1.52 ± 7.16 (0.86, 2.18)	6.46 ± 11.36 ^1^ (5.53, 7.38)	−1.44 ± 6.39 (−2, −0.88)	5.15 ± 16.04 ^1^ (4.16, 6.14)	−5.95 ± 9.59 (−6.96, −4.95)	3.28 ± 17.18 ^1^ (2.16, 4.4)	−25.90 ± 31.62 (−29.46, −22.34)	−9.70 ± 33.85 ^1^ (−11.72, −7.68)
hs-CRP change	−0.041 ± 0.36 (−0.08, 0)	0.004 ± 0.33 (−0.03, 0.04)	−0.018 ± 0.27 (−0.05, 0.01)	0.005 ± 0.48 (−0.03, 0.04)	−0.040 ± 0.18 (−0.07, −0.01)	−0.001 ± 0.24 ^1^ (−0.02, 0.02)	−0.083 ± 0.27 (−0.12, −0.04)	0.009 ± 0.48 ^1^ (−0.03, 0.05)

Data are mean ± standard deviation. Change was calculated as BMI_follow-up_−BMI_baseline_. ^1^
*p* < 0.05 between individuals who resolved and persisted, using independent *t*-tests ALT = alanine aminotransferase, AST = aspartate transaminase, BMI = body mass index, HDL-C = high-density lipoprotein cholesterol, hs-CRP = high-sensitivity c-reactive protein, LDL-C = low-density lipoprotein cholesterol, TG = triglyceride.

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
