# Peer review of "Utility of ALT Concentration in Men and Women with Nonalcoholic Fatty Liver Disease: Cohort Study"

_jcm, 2019, doi:10.3390/jcm8040445_

Reviewer 1 Report

This study analyses a large cohort of data and does hold a lot of potentials to draw significant conclusions. The authors have a commendable job in collecting the data and discussing the limitation of their manuscript. Nonetheless, there are some significant flaws in the presentation of data.

 1.       In the results mentioned about the sex-specific percentage of fatty livers but I could not find that data in the manuscript.

2.       In the result section, the authors mentioned Figure 1A and 1B but the labeling or data for Figure 1 is missing. Figure 1 description as Figure1A and Figure 2B

3.       Again in Figure 2, the authors discussed the probability of fatty liver resolution among the high quartile ALT population but the data is either missing or not clear.

4.       Figure 2 missing Y-axis label

5.       The table blows figure 2 is difficult to interpret. Need a better figure legend or figure description.

6.       Under discussion, the authors concluded that In our population with ultrasound-diagnosed fatty liver …ALT concentration (line 187-189). There is no data supporting that conclusion. 

Author Response

Response to Reviewers

Thank you for carefully reviewing our manuscript. We have incorporated suggested comments and have conducted a new analysis to identify sex-specific ALT cut-points for fatty liver in our population. Specific responses are noted below.

Track changes were used to highlight changes.

We received a waiver for informed consent as de-identified data were retrieved restrospectively. 

 Reviewer 1:

This study analyses a large cohort of data and does hold a lot of potentials to draw significant conclusions. The authors have a commendable job in collecting the data and discussing the limitations of their manuscript. Nonetheless, there are some significant flaws in the presentation of data.

-We greatly appreciate the reviewer’s thoughtful comments. Below are specific responses to your comments.

In the results mentioned about the sex-specific percentage of fatty livers but I could not find that data in the manuscript.

-Thank you for noticing this error. Figure 1 has been corrected to illustrate prevalence of fatty liver in men (a) and women (b).

In the result section, the authors mentioned Figure 1A and 1B but the labeling of data for Figure 1 is missing. Figure 1 description as Figure 1A and Figure 1B.

-Thank you for noticing this error. This has been corrected.

Again in Figure 2, the authors discussed the probability of fatty liver resolution among the high quartile ALT population but the date is either missing or not clear.

-Figure 2 has been revised.

Figure 2 missing Y-axis label.

-Figure 2 has been revised.

The table below Figure 2 is difficult to interpret. Need a better figure legend or figure description.

-Figure 2 has been revised.

Under discussion, the authors concluded that “In our population with ultrasound-diagnosed fatty liver… (line 187-189). There is no data supporting that conclusion.

-We were not clear in our language. We have amended the sentence to reflect that half (referring to those in quartile 1 and 2) had ALT<34 in men and <21 in women. (Page 9).

Reviewer 2 Report

Manuscript Review Comments

Title: “Utility of ALT Concentration in Men and Women with Nonalcoholic Fatty Liver Disease: Cohort Study” (JCM-459662)

 The authors present a prospective study based on a population analysis on ALT values in patients with NAFLD. The study seems interesting and well-conducted. However, I find several shortcomings in the manuscript that should be addressed.

 Comment to the authors

 General comments

-          I think that the study would beneficiate of a test for sensibility and specificity for ALT comparing it to ultrasound examination. This may bring more information about ideal cutoff values for ALT for NAFLD diagnosis.

 Specific comments:

Title page and abstract:

-          Please avoid when possible the use of acronyms in the abstract.

-          Quartiles in the abstract are not specified.

-          Please ensure that the keywords are correct MeSH terms. Use the appropriate terms as specified in MeSH for better indexation.

Materials and Methods

-          How was alcohol consumption assessed in order to exclude potential subjects from the study?

-          The description of the final sample enrolled after applying inclusion and exclusion criteria should be reported at the beginning of the results section.

-          Please comment on the study design and follow an appropriate guideline for the manuscript preparation and specify this in the methods section. In this case, STROBE guidelines are the appropriate for observational studies.

-          State the revision of the Declaration of Helsinki that was used. Also provide a proper approval reference number of the Institutional Review Board.

-          How were defined the physical activity groups?

-          Why was waist circumference not evaluated? It is an importer cardiovascular and metabolic marker.

-          Some of the analyzed biomarkers are not detailed in methods section (Glucose, lipid profile, CRP etc.).

-          How was LDL calculated? Author do not report total cholesterol (Friedewald formula would not be applicable).

-          LDL is referred as LDL-C but HDL is not? Please be consistent in the acronym use.

-          Please state which criteria were applied to define obesity, prediabetes etc. (table 1)

-          Please state the name of the laboratory where the samples were analyzed.

-          Statistical analysis: What test was used for normality test? Which transformations or nonparametric tests were used in each case?

Results

-          Both results and discussion have the heading from the template. Please remove them.

-          There is a great disbalance in terms of gender in the study groups (around 6 times more men).

-          Which statistical tests were applied in table 3 and 4?

 Discussion

-          As mentioned previously, it is a cross-sectional study and hence the results may only be discussed in terms of association of the variables. Authors should address the limitations of this design in the discussion and lower the tone of the conclusions they draw from the results.

Author Response

Response to Reviewers

Thank you for carefully reviewing our manuscript. We have incorporated suggested comments and have conducted a new analysis to identify sex-specific ALT cut-points for fatty liver in our population. Specific responses are noted below.

Track changes were used to highlight changes.

We received a waiver for informed consent as de-identified data were retrieved restrospectively. 

 Reviewer 2:

We greatly appreciated the Reviewer’s thorough review of our manuscript and helpful comments. Specific responses are noted below comments.

 General Comments

-I think that the study would beneficiate of a test for sensibility and specificity for ALT comparing it to ultrasound examination. This may bring more information about ideal cutoff values for ALT for NAFLD diagnosis.

     We have included a sensitivity and specificity analyses for ALT to predict fatty liver at baseline. This information has been included in the Results. See page 3.

AUROC*

ALT

(cut off value)

sensitivity

specificity

total

82.3%

24

73.6%

76.5%

male

78.0%

27

70.6%

71.5%

female

74.8%

19

60.0%

77.3%

AUROC, area under the receiver operating characteristic curve

 Specific Comments

 Title page and abstract

-Please avoid when possible the use of acronyms in the abstract.

     We have eliminated the acronym ULN to limit use of acronyms.

 -Quartiles in the abstract are not specified.

     The abstract was amended to include specific ALT quartile cut-points for men (Q1 5-24, Q2 25-33, Q3 34-48, Q4 ≥49 IU/L) and women (Q1 5-14, Q2 15-20, Q3 21-28, Q4 ≥29 IU/L).

 -Please ensure that the keywords are correct MeSH terms. Use the appropriate terms as specified in MeSH for better indexation.

     Only keywords that are MeSH terms are now included.

 Materials and Methods

-How was alcohol consumption assessed in order to exclude potential subjects from the study?

     Individuals complete a self-administered questionnaire which queried frequency of alcohol intake and quantity of intake when drinking; this information was used to calculate consumption of alcohol per day (grams/day). The authors have previously used this methodology to assess association between alcohol intake and cardiovascular disease (PMID 17623829). We have elaborated on our methodology in the manuscript (page 2, line 81).

 -The description of the final sample enrolled after applying inclusion and exclusion criteria should be reported at the beginning of the results section.

     This information has been added to page 3.

 -Please comment on the study design and follow an appropriate guideline for the manuscript preparation and specify this in the methods section. In this case, STROBE guidelines are the appropriate for observational studies.

     Details related to the study design have now been added to beginning of the Methods section.

 -State the revision of the Declaration of Helsinki that was used. Also provide a proper approval reference number of the Institutional Review Board.

     We analyze de-identified data and thus did not need to obtain informed consent. The study was approved by the Institutional Review Board of Kangbuk Samsung Hospital (KBSMC 2013-01-010). This information has been added to page 2.

 -How were defined the physical activity groups?

     Individuals were asked about frequency of moderate- or vigorous-intensity physical activity per week by asking how many days per week they participated in a sweat-inducing exercise for more than 30 minutes. We have used the criteria<1 time per week vs ≥1 time per week in most of our studies as it delineates individuals who do not exercise vs exercise.

 -Why was waist circumference not evaluated? It is an importer cardiovascular and metabolic marker.

     Unfortunately, waist circumference measurement were missing at baseline in 40% of individuals. We have included this detail in page 2.

 -Some of the analyzed biomarkers are not detailed in methods section (glucose, lipid profile, CRP, etc).

     We apologize for this error. Details have been included on page 3.

 -How was LDL calculated? Author do not report total cholesterol (Friedewald formula would not be applicable).

     LDL was directed measured. Details are noted on page 3.

 -LDL is referred to as LDL-C but HDL is not? Please be consistent in the acronym use.

     Thank you for identifying this inconsistence. This has been corrected throughout the manuscript.

 -Please state which criteria were applied to define obesity, prediabetes, etc. (Table 1).

     Obesity was defined as having BMI ≥ 25 kg/m2; prediabetes, fasting glucose 100-125 mg/dL; diabetes, fasting glucose ≥ 126 mg/dL, hypertension, systolic blood pressure ≥ 140 mmHg and/or diastolic blood pressure ≥ 90 mmHg; CVD, self-reported coronary artery disease and stroke. This information has been added to the bottom of Tables 1 and 2.

 -Please state the name of the laboratory where the samples were analyzed.

     Blood samples were analyzed in the clinical laboratory at Kangbuk Samsung Hospital. This information has been included in the Methods on page 2.

-Statistical analysis. What test was used for normality test? Which transformations or nonparametric tests were used in each case?

     Normal distribution of continuous variables was evaluated using graphical methods to check for skewness and kurtosis. Nonparametric variables (insulin, triglyceride, AST, ALT, hs-CRP) are reported as median and interquartile range (IQR), and logarithmically transformed before analyses. This detail has been added to page 3.

 Results

-Both results and discussion have the heading from the template. Please remove them.

     We have removed the instructions. Thank you.  

-There is a great disbalance in terms of gender in the study groups (around 6 times more men).

     We have more men than women as the screening program is generally offered through work. We have added this as a limitation on page 11.

-Which statistical tests were applied in tables 3 and 4?

     We evaluated for normal distribution of the change values using graphical methods. As change values were symmetrically distributed around 0, we compared differences between groups using independent t-test. This information has been added to Tables 3 and 4.

 Discussion

-As mentioned previously, it is a cross-sectional study and hence the results may only be discussed in terms of association of the variables. Authors should address the limitation of the designe in the discussion and lower the tone of the conclusions they draw from the results.

Thank you for this suggestion. We have added this limitation and modified our conclusion. The last sentence of our conclusion now reads:  “While our study needs to be interpreted with caution given its retrospective design, we suggest that a higher ULN may yield better case finding of those with fatty liver who deserve the greatest clinical attention.

 Round  2

Reviewer 1 Report

The authors have addressed all my comments. 

Reviewer 2 Report

Authors have correctly addressed my previous comments.